# Bio-Guided Isolation of Compounds from *Fraxinus excelsior* Leaves with Anti-Inflammatory Activity

**DOI:** 10.3390/ijms24043750

**Published:** 2023-02-13

**Authors:** Małgorzata Kołtun-Jasion, Paulina Sawulska, Andrzej Patyra, Marta Woźniak, Marta Katarzyna Dudek, Agnieszka Filipek, Anna Karolina Kiss

**Affiliations:** 1Department of Pharmaceutical Biology, Medical University of Warsaw, Banacha 1, 02-097 Warsaw, Poland; 2Doctoral School, Medical University of Warsaw, Żwirki i Wigury 81, 02-091 Warsaw, Poland; 3Institut des Biomolécules Max Mousseron, Université de Montpellier, CNRS, ENSCM, 34293 Montpellier, France; 4Structural Studies Department, Centre of Molecular and Macromolecular Studies, Polish Academy of Sciences, Sienkiewicza H. 112, 90-001 Łódź, Poland

**Keywords:** *Fraxinus excelsior*, ash leaves, inflammation, *Oleaceae*, secoiridoids, phenylethanoids, flavonoids, cytokines, IL-10 receptor, monocytes, macrophages

## Abstract

Inflammation is the first physiological defence mechanism against external and internal stimuli. The prolonged or inappropriate response of the immune system may lead to the persistent inflammatory response that can potentially become a basis for chronic diseases e.g., asthma, type II diabetes or cancer. An important role in the alleviation of inflammatory processes, as an adjunct to traditional pharmacological therapy, is attributed to phytotherapy, especially to raw materials with a long tradition of use, e.g., ash leaves. Despite their long-term use in phytotherapy, the specific mechanisms of action have not been confirmed in a sufficient number of biological or clinical studies. The aim of the study is a detailed phytochemical analysis of infusion and its fractions, isolation of pure compounds from the leaves of *Fraxinus excelsior* and evaluation of their effect on the secretion of anti-inflammatory cytokines (TNF-α, IL-6) and IL-10 receptor expression in an in vitro model of monocyte/macrophage cells isolated from peripheral blood. Methods: Phytochemical analysis was carried out by the UHPLC-DAD-ESI-MS/MS method. Monocytes/macrophages were isolated from human peripheral blood using density gradient centrifugation on Pancoll. After 24 h incubation with tested fractions/subfractions and pure compounds, cells or their supernatants were studied, respectively, on IL-10 receptor expression by flow cytometry and IL-6, TNF-α, IL-1β secretion by the ELISA test. Results were presented with respect to *Lipopolysaccharide* (LPS) control and positive control with dexamethasone. Results: The infusion, 20% and 50% methanolic fractions and their subfractions, as well as their dominating compounds, e.g., ligstroside, formoside and oleoacteoside isolated from the leaves, show the ability to increase the IL-10 receptor expression on the surface of monocyte/macrophage cells, stimulated by LPS, and to decrease the secretion of pro-inflammatory cytokines, e.g., TNF-α, IL-6.

## 1. Introduction

The Fraxinus genus (Oleaceae family) native range is in northern temperate Eurasia and America, and according to the Kew Garden database (https://powo.science.kew.org/ (accessed on 1 December 2022)), it includes 58 accepted species. Many of them are economically important and are grown for timber. *Fraxinus ornus* L. is a commercial source of manna, while some species are used as medicinal plants in traditional Chinese medicine (TCM) and European medicine. In the 10th edition of the European pharmacopoeia, two monographs are presented: *Fraxini chinensis* cortex standardised on coumarins and *Fraxinis folium* standardised on hydroxycinnamic acid derivatives. Ash leaf (*Fraxini folium*) is obtained from *Fraxinus excelsior* L. or *Fraxinus angustifolia* Vahl (syn. *Fraxinus oxyphylla* M. Bieb) or the hybrids or mixture of these two species. However, the occurrence of both species in Europe only partly overlaps. The narrow-leaved ash (*F. angustifolia*) distribution covers central-southern Europe and northwest Africa. Common ash (*Fraxinus excelsior*) is naturally found throughout the European temperate zone and has a wider distribution than the two other native ash species, narrow-leafed ash and manna ash (*Fraxinus ornus* L.) [1,2]. During collection from trees, the distinction between all three species was based on the presence of corolla (*F. ornus*) and the presence of black (*F. excelsior*) or brown (*F. angustifolia*) buds [3].

According to the European Medicines Agency, in European traditional medicine, ash leaves are recommended (1) to treat minor articular pain and (2) to increase the amount of urine for flushing in minor urinary complaints. However, from the time of Hippocrates, ash leaves have been used in several diseases connected with the inflammatory state such as rheumatism, arthritis, gout, neuralgia, kidney and urinary tract diseases; as anti-catarrhal, anti-fever, anti-influenza agents; and as an external treatment for infected or delayed healing wounds, acne and dermatitis [4,5,6,7,8,9,10,11,12,13].

Although on the European market, raw plant material, as well as powdered leaves in capsules are accessible, the number of biological and pharmacological investigations proving the usefulness of this plant material in recommended traditional uses is surprisingly scarce. In the literature, an antioxidative activity based on the inhibition of NADH oxidase activity and reactive oxygen species (ROS) production and ROS scavenging activity was described [14,15,16,17]. In our previous study, we observed that infusions of ash leaves from different sources were able to reduce the in vitro pro-inflammatory functions of neutrophils, such as interleukin 8 (IL-8), interleukin 1β (IL-1β), monocyte chemoattractant protein 1 (MCP-1) and tumour necrosis factor (TNF-α) release [16]. Noteworthy, it was proven that the ethanolic extract of leaves of *Fraxinus angustifolia* showed significant antimutagenic and antigenotoxic effects in *Salmonella typhimurium* TA98 and TA100 strains assay and in an umu test in *S. typhimurium* TA1535/pSK1002 [18].

The pharmacological effect of this plant material is generally attributed to the complex action of flavonoids, phenylethanoids and secoiridoids [19]. However, this was never proven experimentally. The presence of secoiridoids, phenylethanoids, flavonoids and coumarins is a characteristic feature of the *Fraxinus* species, especially the occurrence of coumarins that distinguish the genus *Fraxinus* from the other genera in Oleaceae. In *Fraxinus excelsior*, almost 15 different coumarins and their glycosides were identified in the bark, while leaves did not contain coumarins [16,19,20]. The secoiridoids present in this genera are typical of the Oleaceae family: oleuropein and ligstroside, along with the less common excelsioside and dimer GI5 [16,20,21]. Verbascoside/acteoside is the main phenylethanoid and rutoside is the main flavonoid in leaves [16,19].

In order to elucidate which compounds present in this plant material may be responsible for the anti-inflammatory activity of ash leaves, prepared traditionally as infusion, we used the human monocyte model. Although monocytes make up only 10% of peripheral leucocytes in human blood, they are capable of rapid mobilisation in large numbers to sites of inflammation [22]. These haematopoietic cells involved in both innate and adaptive immunity, after circulation in the peripheral blood, migrate to tissues and undergo classical or alternative activation and eventual differentiation into phagocytic macrophage cells of pro-inflammatory (M1) or anti-inflammatory type (M2) [23]. Monocytes orchestrate the innate immune response to LPS by expressing a variety of inflammatory cytokines: tumour necrosis factor-alpha (TNF-α), IL-6, IL-1β, G-CSG, GM-CSF, M-CSF and chemokines: IL-8, MCP-1 [24]. The release of pro-inflammatory cytokines and the recruitment of inflammatory cells at the site of injury lead to the cytokine storm phenomenon. Thus, the mechanisms controlling the movement of monocytes under conditions of homeostasis, infection and inflammation are still being discovered, and the search for substances that control ongoing inflammatory processes arouses keen interest in the research community. Furthermore, the controlled inflammatory process is an essential mechanism for promoting organism balance. IL-10, secreted by monocytes and macrophage (M2) cells, keeps pro- and anti-inflammatory events under control, protecting against excessive immune responses and tissue damage [25]. Although the molecular basis underlying IL-10 functional redundancy and plasticity is still challenging for researchers, it may be considered an exquisite immune mediator. The activation of the cell surface IL-10 receptor complex is the first step in initiating IL-10 signalling pathways. Impaired IL-10R function is associated with the pathology of many diseases and chronic immunodeficiency and autoimmune states, including allergies, inflammatory-based infections or rheumatoid arthritis [26,27,28,29]. 

Ash leaf preparations, due to the partly confirmed anti-inflammatory effects and relative safety of use, could be used chronically and for a long time as an addition to traditional pharmacotherapy. Despite their availability, they are relatively popular in the European market for use mostly as self-made tea (infusion). The aim of the present study is therefore to investigate the effects of the infusion, its fractions and, finally, the biopreparations isolated on pro- and anti-inflammatory functions of monocytes, such as the release of interleukin 6 (IL-6), interleukin 1β (IL-1β) and tumour necrosis factor (TNF-α) and their effect on IL-10 receptor expression. The obtained results allowed us to confirm the hypothesis of a pleiotropic effect of the studied medicinal plant, both in the context of inhibition of pro-inflammatory cytokine secretion and on the impact on the anti-inflammatory IL-10 receptor expression. The active compounds dominating in the ash leaves may serve to standardize commercial ash leaf preparations and provide leading structures for further research into the use of ash leaves in therapy and prevention.

## 2. Results

### 2.1. Phytochemical Characterisation of Ash Leaves Infusion 

The phytochemical analysis of ash leaves infusion was performed using the HPLC-DAD-MS/MS method. In a previous study, we identified or partly identified 68 compounds which may be present in *Fraxini folium* [16]. However, in the infusion, the most abundant were chlorogenic and neochlorogenic acids (3-*O*-caffeoylquinic and 5-*O*-caffeoylquinic acids), the flavonoids-rutin (quercetin-3-*O*-rutinoside) and a kaempferol derivative, the phenylethanoid-acteoside, and secoiridoids-oleuropein, formoside, ligstroside and a phenylethanoid esterified with an oleoside 11-methyl ester-oleoacteoside (Figure 1 and Table 1). 

### 2.2. Effect of Ash Leaves Infusion and Its Fractions on the Proinflammatory Function of LPS-Stimulated Monocytes

The crude lyophilised infusion and its fractions were obtained by using SPE RP-18 column and water (Extract H_2_O), 20% methanol (Extract 20%), 50% methanol (Extract 50%), 70% methanol (Extract 70%) and 100% methanol (Extract 100%) as eluents, and they were tested using the monocyte model of LPS-induced inflammation. The activation of monocyte cells by LPS resulted in an induction of the release of cytokines such as IL-6, TNF-α and IL-1β, in comparison to the untreated control (Figure 2A–C). Incubation for 24 h of LPS-stimulated cells with the infusion and fractions from *Fraxinus leaves* (50 μg/mL) resulted in a reduction of proinflammatory cytokine release, such as IL-6, TNF-α, without a significant effect on the IL-1β release. The effect was statistically significant in the case of the crude infusion and two extracts obtained with 20% and 50% methanol elution (*p* < 0.01). Infusion and 50% methanolic extracts reduced the TNF-α release to a comparable degree to the positive control with dexamethasone (20 μM). Moreover, the IL-10/IL-10R pathway plays a critical role in the control of immune responses and the regulation of tissue homeostasis. Recent studies confirm macrophages as being the main targets of the inhibitory IL-10 cell effects. Interestingly, in the case of the conducted study, the infusion itself and the 50% methanolic extract were only inactive (Figure 2D).

At the same time, the potential cytotoxic effect was also investigated using LDH assay. At the tested concentration, no significant diminished membrane integrity in comparison with controls was observed. 

### 2.3. Bio-Guided Isolation of Active Compounds from 20% and 50% Extracts Obtained from Ash Leaves Infusion

The most active extracts 20% and 50% were analysed using HPLC-DAD-MS/MS methods and revealed the domination of phenylethanoids and secoiridoids. However, the presence of minor compounds such as flavonoids was also observed. 

Further fractionation using a Sephadex LH-20 column allowed us to obtain, based on their TLC and HPLC profiles, nine main fractions (F1–F9_20%) from Extract 20%, and seven main fractions (F1–F7_50%) from Extract 50%. The bioactivity of fractions was evaluated by determining the inhibition of IL-6, TNF-α and IL-1β release (Figure 3A–C). Fraction F3_20% from the 20% extract shows a significant inhibition of TNF-α release and exhibits a tendency to decrease the IL-6 release. In turn, the F7_20% fraction showed a statistically significant inhibitory effect on IL-1β secretion compared to the LPS-stimulated control.

The phytochemical analysis allowed the characterisation of a dominating compound from fraction F3_20% as a ligstroside derivative (tr = 39.6 min, a pseudomolecular ion (569 [M-H+HCOOH]^−^ and fragment ions at *m*/*z* 523 [M-H]^−^ and *m*/*z* 361 [M-H-162]^−^) and from fraction F7_20%, verbascoside/acteoside was identified based on comparison with a standard. Interestingly, from the 50% methanolic extract, fraction F7_50% decreased only IL-6 release, and F5_50% the TNF-α release, while the F2_50% fraction contributed to the inhibition of IL-1ß release. (Figure 3A–C). Fraction F2_50% was enriched with one dominating compound, ligstroside derivative (tr = 39.6 min, a pseudomolecular ion (569 [M-H+HCOOH]^−^). Similarly to the fraction F3_20%, fraction F5_50% contained mostly quercetin rutoside, verbascoside and kaempferol hexosylrhamnoside, while fraction F7_50% was rich in flavonoids, aglycones, quercetin and kaempferol. However, the last two compounds are minor constituents of the infusion. 

Moreover, fractions from Extract 20% were also analysed for their ability to induce IL-10 receptor expression, and interestingly in this case, fractions F1_20%, F5_20% and F6_20% were active, suggesting that different compounds are responsible for the pleiotropic anti-inflammatory activity of ash leaves (Figure 4). Fraction F1_20% did not contain any secondary metabolites and probably may contain some polysaccharides, while F5_20% and F6_20% were enriched by secoiridoids, oleuropein and ligtroside, and a phenylethanoid esterified with an oleoside 11-methyl ester was tentatively identified as oleoacteoside (tr = 39.9 min; pseudomolecular ion [M-H]^−^ at *m*/*z* 1009 and fragment ions at *m*/*z* 847 [M-H-162]^−^, *m*/*z* 623 [M-H-162-224]^−^ and *m*/*z* 461 [M-H-162-224-162]^−^, corresponding to the cleavage of hexose, elenolic acid and caffeoyl moieties). 

From the fractions F3_20% and F5_20%, using Sephadex LH-20 chromatography and preparative chromatography, three compounds were isolated (Figure 5). Their structures were confirmed by 1H and 13C NMR spectra by comparison with reference data [21,30,31].

### 2.4. Bioactivity of Compounds Present in the Most Active Fractions

In order to elucidate which compounds are responsible for the activity of the infusion, we tested compounds which were identified and isolated from the active fraction at the concentration of 20 µM. All compounds were active in reducing TNF-α release, while oleuropein, ligstroside, verbascoside and rutoside at concentrations 20 µM were able to significantly and comparably to positive control with dexamethasone (20 µM), decrease the IL-6 release from stimulated monocytes. Only formoside and oleoacteoside had the ability to reduce IL-1β release. The main representative of secoiridoids in this raw material, oleuropein, as well as phenylpropanoid derivatives, increased the expression of the IL-10 receptor to a clinically relevant extent (Figure 6). 

## 3. Discussion

Monocytes are haematopoietic cells involved in both innate and adaptive immunity. Circulating in the peripheral blood, monocytes migrate to tissues and undergo classical or alternative activation and eventual differentiation into phagocytic macrophage cells of pro-inflammatory (M1) or anti-inflammatory type (M2). The rapid and massive overproduction of inflammatory mediators to the presence of LPS can lead to sepsis, septic shock or systemic inflammatory response syndrome, which is why it is so important to suppress ongoing inflammation in its early stages [24]. Taking into account the role of chronic inflammation underlying metabolic diseases (i.e., type II diabetes, hypertension or atherosclerosis) and other chronic diseases (including chronic obstructive pulmonary disease or Alzheimer’s disease), the search for new methods to prevent their development seems indispensable. Furthermore, shedding light on the changes occurring in immune cells, i.e., monocytes, macrophages or neutrophils, in the development of inflammatory-based diseases, their candidature should be highlighted as potential therapeutic targets [32]. For this reason, preparations of natural origin are attracting increasing interest as adjuncts to conventional therapies for inflammatory disorders, as well as methods of disease prevention.

Our study aimed to assess the relative importance of ash leaves as an ingredient in anti-inflammatory preparations used in the treatment of articular, urological and dermatological conditions, according to the indications contained in the EMA assessment. Moreover, because of the long-standing tradition of using ash leaves, the efficacy of which has not been confirmed in clinical studies, we attempted to evaluate the mechanisms of its action at the cellular level. In addition, a detailed phytochemical analysis of the infusion and methanol extracts from the raw material, as well as the determination of their biological activity towards anti-inflammatory activity allowed for the selection of compounds potentially responsible for the anti-inflammatory properties of ash leaves. Previous studies of ash preparations carried out by Kiss et al. [16] in a human neutrophil cell model confirmed the inhibitory effect of ash leaves on the secretion of pro-inflammatory cytokines (IL-1β, TNF-α, IL-8) and chemokines (MCP-1), as well as on ROS production. In our study, we adopted as a research model the most principal immune effector cells—monocytes isolated from human peripheral blood. Monocyte recruitment is essential for the effective control of viral, bacterial, fungal and protozoal infections, but recruited monocytes also contribute to the pathogenesis of inflammatory and degenerative diseases [23]. To the best of our knowledge, this is one of the first reports on the anti-inflammatory effect of ash leaf preparations on this cell model. Moreover, the confirmation of the anti-inflammatory effect of ash leaves, expressed by the increase in IL-10 receptor expression, as well as the inhibition of the secretion of pro-inflammatory cytokines (IL-1β, TNF-α, IL-6), places this medicinal plant in a favourable light for further clinical considerations in the search for its new applications.

To gain insight into the mechanism of action of the ash leaves, on the third day after isolation, mononuclear cells were stimulated with LPS, followed by treatment with the tested infusion/methanol extracts or single compounds. It was observed that both the infusion and 20% and 50% methanolic extracts, derived from the fractionation of infusion, significantly decreased the pro-inflammatory cytokines (TNF-α, IL-6) release compared to the control stimulated only by LPS. Interestingly, TNF-α secretion was inhibited to a degree similar to the level of inhibition obtained with dexamethasone in the same concentration, taken as a positive control. In contrast, all tested compounds (oleuropein, ligstroside, formoside, oleoacteoside, verbascoside, rutoside) identified or isolated from the most active fractions moderately suppressed the LPS-stimulated release of TNF-α to a greater extent than individual subfractions of the extracts. 

The reduction in IL-1β secretion further contributes to a reduction in its autocrine ability to stimulate the synthesis of other cytokines (e.g., TNF-α, IL-8) and ROS [33]. Oleoacteoside and verbascoside showed the greatest activity in inhibiting IL-1β secretion, while not affecting IL-6. However, the observed effect was less pronounced than with formoside. The observed results regarding pro-inflammatory cytokines release may indicate a synergistic action of the active compounds in obtaining the final anti-inflammatory effect of the tested raw material. To our knowledge, this is one of the few pieces of information concerning the isolation and determination of the biological activity of an oleoacteoside that has previously been isolated from the *Syringa* genus [34]. 

An interesting case concerns oleuropein, the major secoiridoid compound of the *Oleaceae* family that exhibits a wide range of antioxidant, anti-inflammatory, anti-diabetic, neuro- and cardio-protective, anti-microbial and immunomodulatory activities [35]. The results of our study are consistent with those reported by Cui et al., in which oleuropein significantly reduced the level of inflammatory factors TNF-α and IL-6, increased by LPS treatment, via the classic NF-κB and MAPK inflammatory pathways [36]. On the other hand, our study did not show statistically significant inhibition of IL-1β secretion under the influence of oleuropein, which could have been related to the concentration of the tested compound. In another study in a chondrocyte model, oleuropein concentrations of 100 µM showed statistically significant inhibition of NF-κB and MAPK pathway activation in the presence of IL-1β [37]. On the other hand, an in vivo study showed significant anti-inflammatory activity of oleuropein (100 mg/kg food) on the expression of the NLRP-3 inflammasome complex on LPS-stimulated peritoneal macrophages isolated from induced mice, clearly indicating the need to use higher concentrations of the compound to induce a statistically significant anti-inflammatory effect in regard to the IL-1β pathway [38]. Another study conducted by Cui Y. et al. demonstrated the protective effect of high doses of oleuropein (10–40 mg/kg) in an LPS-stimulated renal inflammation mice model and its significant effect on renal parameters, confirming the need for higher concentrations of this compound to achieve full protective effects [36].

IL-10, due to its pleiotropic and seemingly contradictory properties, plays a key role in the oversight of the body’s immune response. IL-10 production can be mediated through negative feedback of pro-inflammatory stimuli to control inflammation. By keeping pro- and anti-inflammatory events under control, it protects against excessive immune responses and tissue damage, maintaining immune system balance. IL-10 action decreases various aspects of inflammation such as fever, hyperalgesia, acute phase protein release and vascular permeability [39,40]. 

Zigmond et al. showed that IL-10Rα-deficient macrophages secrete several pro-inflammatory cytokines that initiate cell response, as indicated by elevated serum titres of IL-17 and IL-6 [41]. Our study supported the hypothesis that IL-10 signalling in haematopoietic cells is crucial for the control of hyperinflammation. Interestingly, in vivo studies have shown that a loss of IL-10 receptor expression in intestinal macrophages is an intrinsic factor in the development of severe colitis. As shown in in vivo studies, mutations in IL-10 or IL-10R signalling underlie disturbances in intestinal integrity and homeostasis, which can lead to severe forms of IBD in humans, and dysregulated IL-10 function has also been associated with cancer and chronic inflammation [29].

Increased IL-10 production most probably represents a compensatory mechanism due to increased expression of proinflammatory cytokines and is a negative regulator of inflammation, which corroborates our findings. Our results indicated that monocyte cells stimulated by bacterial LPS and treated with ash leaves with 50% methanol extract significantly increased the expression of the receptor for IL-10, as did single compounds isolated from the most active subfractions. Oleuropein, as well as verbascoside and oleoacteoside, two main representatives of phenylethanoids, increased the expression of the receptor for IL-10 by approximately 34%, 18%, 21%, respectively, compared to the LPS control. Interestingly, at the same time, oleuropein and verbascoside at a concentration of 20 µM showed a statistically significant reduction in IL-6 secretion. In the case of IL-10 receptor expression, no statistically significant difference was found between formoside and ligstroside and the control group (*p* > 0.05).

Furthermore, elevated levels of IL-10 have been shown to protect against tissue damage, including limiting diabetic wound formation, through positive effects on vascular permeability [42]. On the other hand, IL-10Rβ activity can also be induced in some non-haematopoietic cells such as skin fibroblasts, keratinocytes and epithelial cells following LPS activation, which may indicate the advisability of the external application of certain preparations of natural origin, including ash leaves, for the treatment of chronic, difficult-to-heal wounds, atopic eczema, dermatitis or psoriasis [43,44,45]. 

In chronic urinary tract infections and cystitis, it has been confirmed that urothelial cells, monocytes, macrophages and lymphocytes are a major component of the inflammatory infiltrate in the bladder in response to *E. coli* infection. It is worth noting that IL-10 has well-defined roles in mediating the pathogenesis of bacterial infections due to its broad immunosuppressive effects. A study in patients with uropathogenic Escherichia coli (UPEC) infections showed the presence of significant amounts of IL-10 in the urine and high levels of IL-10 systemically during urosepsis [46]. In contrast, in a study conducted by Jung et al., therapy with macrophages overexpressing the anti-inflammatory factor- interleukin IL-10 had a positive effect on reducing the local inflammatory profile, with a concomitant increase in the expression of pro-regenerative lipocalin-2, protecting against acute kidney injury [47]. Our study confirms a significant increase in the expression of the receptor for IL-10 under the influence of ash leaf preparations, which may imply the advisability of using the raw material in the treatment of urinary tract disorders.

Regardless of the obtained results, further studies on the other cellular models, as well as detailed studies under in vivo conditions, are needed to be able to fully elucidate the mechanism of action of ash leaf preparations, bearing in mind potentially other, still unexplored, molecular mechanisms of action of this raw material.

## 4. Materials and Methods

### 4.1. Chemicals and General Experimental Procedures

Cell culture: LPS (from *Escherichia coli* 0111:B4), HEPES buffer and RPMI 1640 medium with GlutaMAX supplement were purchased from Sigma-Aldrich Chemie GmbH (Steinheim, Germany). Phosphate-buffered saline (PBS) was purchased from Gibco (Grand Island, NY, USA). Ficoll Hypaque gradient (LSM 1077) and penicillin–streptomycin were obtained from PAA, Laboratories GmbH (Pasching, Austria). Human Quantikine ELISA Kits were purchased from the R & D System (Minneapolis, MI, USA). LDH release assay kit was purchased from Roche Applied Science. The absorbance in 96-well microtiter plates was measured using a BioTek microplate reader (Highland Park, Winooski, VT, USA). The flow cytometry was performed using the BD FACSCalibur apparatus (BD Biosciences, San Jose, CA, USA).

Plant material preparation: NMR spectra were registered with a Bruker Avance III 600 spectrometer (Bruker Biospin, Germany) resonating at 600.14 and 150.92 MHz for 1H and 13C, respectively, All measurements were performed at 295 K. All signals were calibrated at solvent residual signals at 3.31 ppm for 1H and 49.50 ppm for 13C (methanol-d4). Preparative HPLC was performed with a Shimadzu LC-20AP instrument (Japan) using a KinetexXB-C18 column (Phenomenex, USA, particle size 5.0 μm, 150 × 21.2 mm) at a flow rate of 20.0 mL/min. TLC was performed on Merck silica gel 60 F 254 (0.25 mm) with dichloromethane/methanol/formic acid/water (80:25:1.5:4; *v*/*v*/*v*/*v*). Chromatograms were visualised by spraying with vanillin in sulfuric acid, followed by heating at 105 °C for 10 min. All solvents used for chromatography were of gradient grade. HPLC-DAD-MSn analysis was performed on a UHPLC-3000 RS system (Dionex, Germering, Germany) with DAD detection and an AmaZon SL ion trap mass spectrometer with an ESI interface (Bruker Daltonik GmbH, Bremen, Germany). Separation was performed on a Zorbax SB-C18 column (150 × 2.1 mm, 1.9 μm) (Agilent, Santa Clara, California, USA). The mobile phase consisted of 0.1% HCOOH in water (A) and 0.1% HCOOH in MeCN (B) using the following gradients: 0–60 min, 5–40% B. The LC eluate was introduced into the ESI interface without splitting, and compounds were analysed in the negative ion modes with the following settings: nebuliser pressure of 40 psi; drying gas flow rate of 9 L/min; nitrogen gas temperature of 300 °C; and a capillary voltage of 4.5 kV. The mass scan ranged from 100 to 2200 *m*/*z*. UV spectra were recorded in the range of 200–400 nm.

### 4.2. Plant Material

Leaves of *Fraxinus excelsior* were collected in June and July 2017 from native plants growing in Warsaw, Mazovian district (52°12′42″ N 21°00′07″ E), Poland. The plant materials were authenticated according to Flora Europaea [3] by Anna K. Kiss. A voucher specimen (no. FE062017) was deposited in the Plant Collection, Department of Pharmacognosy and Molecular Basis of Phytotherapy, Medical University of Warsaw, Poland.

### 4.3. Extracts Preparation, Fractionation and Isolation of Active Compounds

The air-dried leaves (180 g) were crushed, and an infusion was prepared by adding 3 L of boiling water (2×) for 20 min under cover each time. The collected infusion was extracted using solid phase extraction (SPE). Briefly, the infusion (INF) components were absorbed on the RP-18 column (43.5 × 5 cm) and then eluted under reduced pressure with 1.5 L of water, 20% methanol, 50% methanol 70% methanol and 100% methanol, respectively. The methanol from each fraction was evaporated under reduced pressure, and the aqueous residues were lyophilised to obtain five extracts of 12.8 g, 5.3 g, 7.1 g, 1.16 g and 1.64 g, respectively. All fractions were characterised using HPLC-DAD-MS/MS method. The presence of substances in extracts was confirmed by the comparison of retention time and spectra (UV, MS, MS/MS) with standards or/and with literature data. As 20% methanolic and 50% methanolic extracts were shown to be active, both extracts were further separated using bio-guided fractionation. The 20% methanolic extract was subjected to Sephadex LH-20 (Pharmacia) column (85 × 2.5 cm) and eluted with 35% methanol to obtain 80 fractions of 10 mL, which were pooled into 9 main fractions (F1–9_20%) based on their TLC profiles. The 50% methanolic extract was subjected to Sephadex LH-20 (Pharmacia) column (85 × 2.5 cm) and eluted with 35% methanol to obtain 120 fractions of 10 mL, which were pooled into 7 main fractions (F1–7_50%) based on their TLC profiles.

From fraction F3_20% (122 mg), 8E-formoside (6.6 mg; RT = 21.0–22.2 min) was isolated using preparative HPLC with a 0.1% HCOOH in H_2_O (A)-0.1% HCOOH in MeCN (B) gradient (85:15 → 72.5:27.5) in 30 min. From fraction F5_20% (528 mg), oleoacteoside (54 mg; RT = 20.1–21.8 min), and (-)-8E-ligstroside (214 mg; RT = 23.3–26.5 min) were isolated using preparative HPLC using the same condition as for fraction F3_20%.

### 4.4. Preparation of Solutions of Extracts and Compounds for Bioassay

Tested infusion fractions were dissolved in DMSO (10 mg/mL). All test compounds and the positive control dexamethasone were dissolved in DMSO (10 mM stock solution) and then diluted with (Mg^2+^, Ca^2+^)-free PBS buffers at pH 7.4 or RPMI 1640 medium. The infusion and fraction were tested at the concentration of 50 µg/mL; compounds were tested at a concentration of 20 µM. The concentration of DMSO used (<0.1%) did not influence the performed assays.

### 4.5. Isolation of Human Monocytes

Peripheral venous blood was taken from healthy human donors (18–35 years old) in the Warsaw Blood Donation Centre. Donors did not smoke or take any medication. They were clinically recognised to be healthy and routine laboratory tests showed all values to be within the normal ranges. The study conformed with the principles of the Declaration of Helsinki. 

Monocytes were isolated immediately after collection using a Ficoll Hypaque gradient according to Zapolska-Downar et al. [48]. The mononuclear cell band was removed by aspiration, and cells were suspended in RPMI 1640 medium with L-glutamine, HEPES, antibiotics and autologous serum (20%). To allow the adherence of monocytes/macrophages, the peripheral blood mononuclear cell suspension was placed in 12-well tissue culture plates (2 × 10^6^ per well) and incubated for 2 h at 37 °C under humidified 5% CO_2_ air. After this time, non-adherent cells were removed, and adherent cells were cultivated in the same medium and conditions for 3 days. The medium and autologous serum was replaced every 2 days.

### 4.6. Cytotoxicity

Cytotoxicity of the tested infusion, fractions and compounds was determined by using a lactate dehydrogenase (LDH) release assay kit. After 24 h of monocyte cell incubation with the tested infusion, extracts, or compounds, the supernatants were harvested and the LDH level was measured according to the protocol. TRITON X-100 (2%) was used as a positive control. The presence of the exclusively cytosolic enzyme, LDH, in the cell culture medium was calculated referring to the effect of Triton X-100 taken as 100% of cytotoxicity. 

### 4.7. IL-6, IL-1β and TNF-α Secretion

Monocyte cells (2 × 10^6^) were cultured in 12-well plates in RPMI 1640 medium with HEPES, L-glutamine and autologous serum (5%), in the absence or presence of LPS (100 ng/mL) for 24 h at 37 °C with 5% CO_2_, with or without tested infusion/fractions/compounds. After 24 h, the peripheral blood mononuclear cells were harvested and centrifuged (2000 RPM; 10 min; 4 °C), and the supernatant was collected for future experiments. The level of released cytokines was measured by enzyme-linked immunosorbent assay (ELISA) following the manufacturer’s instructions (BD Biosciences, USA). The effect on IL-6, TNF-α and IL-1β production was calculated by comparing the percentages of the released agents to the stimulated control, but untreated with the tested infusion/extracts/compounds. 

### 4.8. Expression of IL-10 Receptor at the Surface of Monocytes/Macrophages

The expression of the IL-10 receptor on the surface of monocytes/macrophage cells was determined by flow cytometry. Cells were incubated with LPS at a concentration of 100 ng/mL for 1 h and then with tested infusion/fractions/extracts/compounds for 24 h. All solutions were added to the cells on the fourth day of incubation. Cells then were removed, centrifuged (13,000 RPM, 4 °C, 1 min), suspended in PBS (100 μL) and incubated with the related antibody (PE Rat Anti-human IL-10, BD Pharmingen, BD Bioscience, USA) for 20 min at 4 °C. The mean fluorescence intensity in the gated cell population was measured (10,000 cells per sample) and analysed by flow cytometry. The results were expressed as the percentage of cells expressing IL-10 receptor in comparison to stimulated control by LPS.

The results were expressed as a mean ± SEM of three independent experiments performed at least in duplicate. All analyses were performed using GraphPad Prism 8.4.3 software. The statistical significance of the differences between means was established by ANOVA with Dunnett’s post hoc test. *p* Values below 0.05 were considered statistically significant

## 5. Conclusions

The observed inhibition of inflammatory cytokine production (TNF-α, IL-6, IL-1β) is of particular importance in the treatment of inflammatory-based diseases. The obtained results demonstrate the anti-inflammatory efficacy of ash leaf formulations and confirm the pharmacopeial indications for the use of this raw material in European medicine. In addition, our study confirms the importance of ash leaves as a valuable source of active compounds, especially oleuropein, ligstroside, and verbascoside, which is essential for further research regarding their use in monocyte activation. Moreover, in future, the synergistic combination of ash leaf preparations with conventional drug therapy may increase the potency of synthetic drugs at lower effective doses, thereby minimising side effects and toxicity.

## Figures and Tables

**Figure 1 ijms-24-03750-f001:**
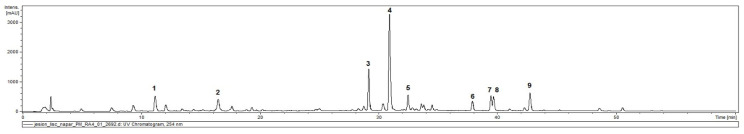
HPLC-DAD chromatograms of the infusion of common ash leaf sample recorded at 254 nm.

**Figure 2 ijms-24-03750-f002:**
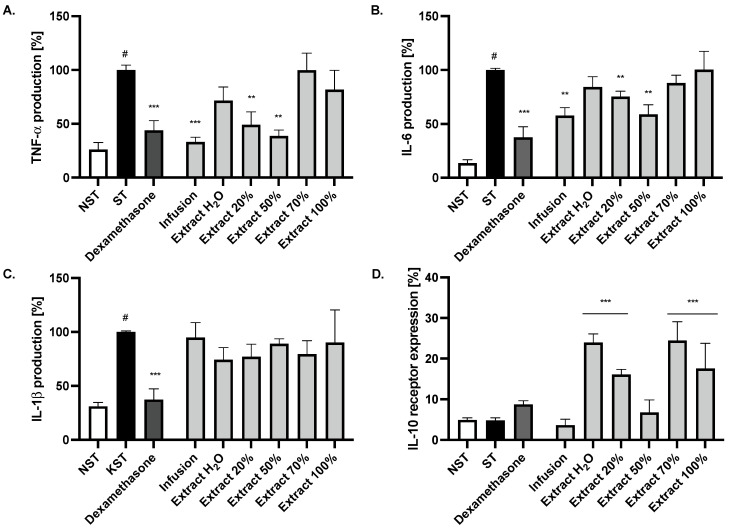
The influence of infusion and selected extracts on (**A**) TNF-α, (**B**) IL-6, (**C**) IL-1β secretion and (**D**) IL-10 receptor expression by LPS-stimulated monocytes/macrophages. Non-stimulated cells (NST), LPS-stimulated (ST) monocytes/macrophages and LPS-stimulated monocytes/macrophages treated with 50 μg/mL of infusion or selected extracts. Dexamethasone (20 μM) was used as a positive control. Data are expressed as mean ± SEM (three separate experiments performed on monocytes/macrophages isolated from independent donors). Statistical significance: ** *p* < 0.01, *** *p* < 0.001 versus stimulated control, # statistically significant (*p* < 0.001) versus non-stimulated control.

**Figure 3 ijms-24-03750-f003:**
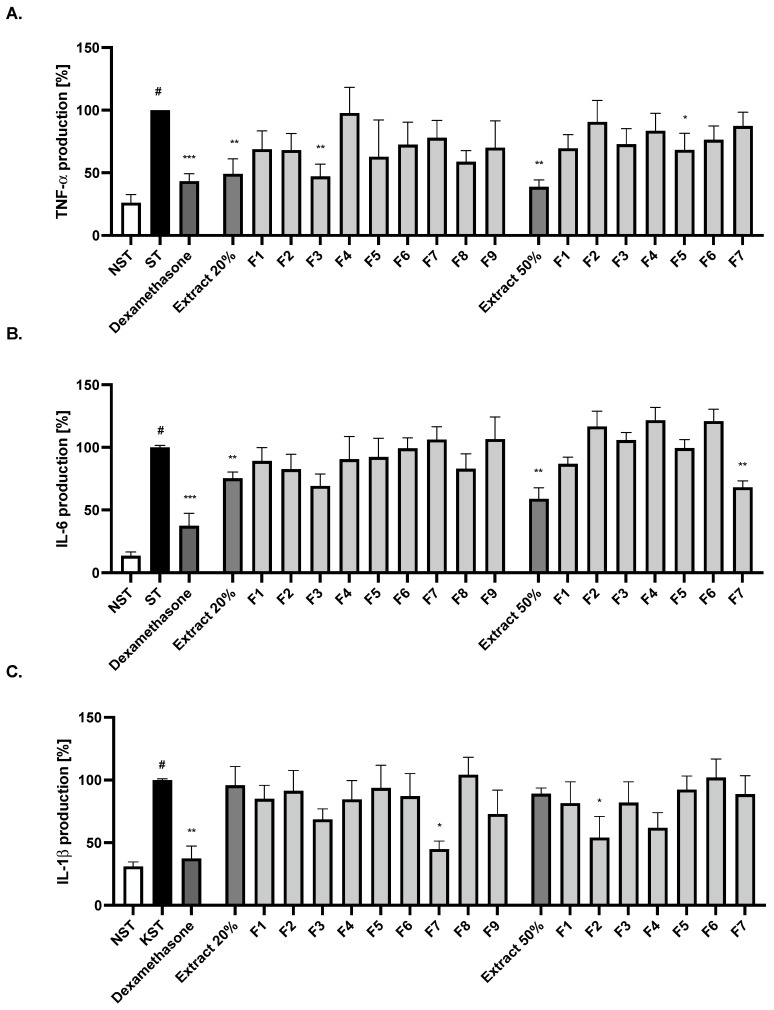
The influence of methanolic extracts and selected fractions on (**A**) TNF-α, (**B**) IL-6 and (**C**) IL-1β production by LPS-stimulated monocytes/macrophages. Non-stimulated cells (NST), LPS-stimulated (ST) monocytes/macrophages and LPS-stimulated monocytes/macrophages treated with 50 μg/mL of infusion, extracts and fractions. Dexamethasone (20 μM) was used as a positive control. Data are expressed as mean ± SEM (three separate experiments performed on monocytes/macrophages isolated from independent donors). Statistical significance: * *p* < 0.05, ** *p* < 0.01, *** *p* < 0.001 versus stimulated control, # statistically significant (*p* < 0.001) versus non-stimulated control.

**Figure 4 ijms-24-03750-f004:**
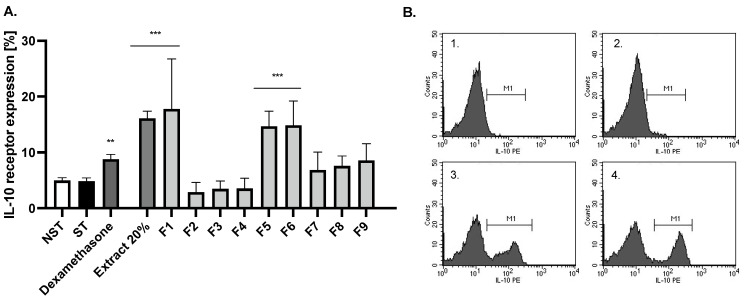
The influence of 20% methanolic extracts on IL-10 receptor expression by LPS-stimulated monocytes/macrophages (**A**). Representative flow cytometry images of IL-10R expression on the surface of monocytes/macrophages (**B**) in non-stimulated control (**1**.), LPS-stimulated control (**2**.), F4 50% MeOH (**3**.), F6 50% MeOH (**4**.) Non-stimulated cells (NST), LPS-stimulated (ST) monocytes/macrophages and LPS-stimulated monocytes/macrophages treated with 50 μg/mL of selected fractions. Dexamethasone (20 μM) was used as positive control. Data are expressed as mean ± SEM (three separate experiments performed on monocytes/macrophages isolated from independent donors). Statistical significance: ** *p* < 0.01, *** *p* < 0.001 versus stimulated control.

**Figure 5 ijms-24-03750-f005:**
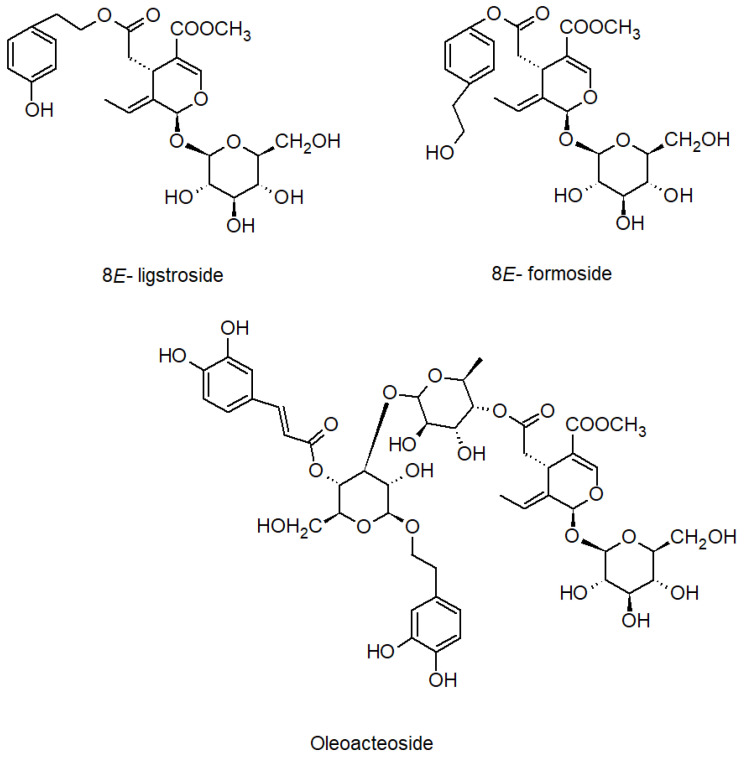
Structure of isolated compounds.

**Figure 6 ijms-24-03750-f006:**
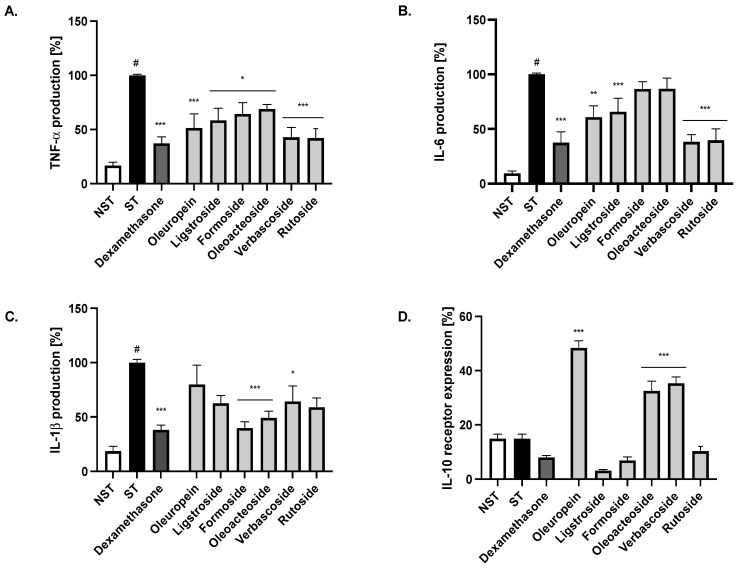
The influence of compounds on (**A**) TNF-α, (**B**) IL-6, (**C**) IL1-β secretion and (**D**) IL-10 receptor expression by LPS-stimulated monocytes/macrophages. Non-stimulated cells (NST), LPS-stimulated (ST) monocytes/macrophages and LPS-stimulated monocytes/macrophages treated with 20 μM of selected compounds. Dexamethasone (20 μM) was used as positive control. Data are expressed as mean ± SEM (three separate experiments performed on monocytes/macrophages isolated from independent donors). Statistical significance: * *p* < 0.05, ** *p* < 0.01, *** *p* < 0.001 versus stimulated control, # statistically significant (*p* < 0.001) versus non-stimulated control.

**Table 1 ijms-24-03750-t001:** Retention time, UV and MS/MS data of compounds identified in ash leaf infusion using UHPLC-DAD-ESI-MS/MS method.

Compounds	UV [nm]	Rt [min]	[M-H]^−^	Product Ions Main Peaks
3-*O*-caffeoylquinic acid	325	11.4	353	191, 179
2. 5-*O*-caffeoylquinic acid ^a^	325	16.6	353	191
3. Quercetin rutinoside ^a^	256, 355	29.2	609	301
4. Verbascoside/acteoside ^a^	215, 330	31.1	623	461, 315, 161, 315
5. Kaempferol hexosylrhamnoside	264, 343	32.7	593	285
6. Oleuropein ^a^	222, 275	38.0	539	377, 307, 275
7. 8*E*-formoside ^b^	218	39.6	569 *	523, 385, 299, 223
8. Oleoacteoside ^a, b^	216, 325	39.9	1009	847, 623, 461
9. 8*E*- ligstroside ^a, b^	216, 280	43.0	523	523, 361

* [M-H+HCOOH]-, ^a^ identified with authentic standards, ^b^ isolated in this study.

## Data Availability

Not applicable.

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
