# Peer review of "Bio-Guided Isolation of Compounds from Fraxinus excelsior Leaves with Anti-Inflammatory Activity"

_ijms, 2023, doi:10.3390/ijms24043750_

Round 1
Reviewer 1 Report
Manuscript described the anti-inflammatory activity of infusion from Fraxinus excelsior and isolated components of the infusion. Introduction gives relevant background and the aim of the study is well justified. The experimental part is well planned and the results are clearly presented. However, I have some suggestions for Authors:
Title could more reflect the study
Abstract: lines 22, 24, 70 lack of italic. Moreover, lines 12-20 – this part should be shortened.
Line 45: the abbreviation “TCM” should be explained
Line 143: „The crude lyophilised infusion and its extracts were” – fractions?
Line 149: the word „fraction” should be more appropriate than „extracts”
Line 145: it should be: „and they were tested”
Line 221-222: „3 compounds were isolated: 8E-formoside (syn. excelsioside), oleoacteoside, and 8E-ligstroside were isolated” – remove it
Figures 2,3 and 5: add the determined factor on y-axis (e.g. TNF-α, IL-6, IL-1β…)
Numbering of Figures should be corrected. Figure 6 should be cited after Figure 5
Figure 6: the name of compounds are hardly to read
Line 416: „The methanol from combined fractions” – were the fraction combined?
Line 437/438: „were tested at the concentration range of 50 μg/mL” - „range” is unnecessary
Line 464: „(2 x 106)” – use the superscript for 6
4.1. Chemicals and general experimental procedures – the paragraph should be divided on subsection to facilitate following
Author Response
Thank you for your remarks,
Abstract: lines 22, 24, 70 lack of italic. Moreover, lines 12-20 – this part should be shortened. – this was done
Line 45: the abbreviation “TCM” should be explained- this was done
Line 143: „The crude lyophilised infusion and its extracts were” – fractions? - this was corrected
Line 149: the word „fraction” should be more appropriate than „extracts”- this was corrected
Line 145: it should be: „and they were tested”-thank you, this was corrected
Line 221-222: „3 compounds were isolated: 8E-formoside (syn. excelsioside), oleoacteoside, and 8E-ligstroside were isolated” – remove it - this was corrected
Figures 2,3 and 5: add the determined factor on y-axis (e.g. TNF-α, IL-6, IL-1β…)- this was done
Numbering of Figures should be corrected. Figure 6 should be cited after Figure 5- this was corrected
Figure 6: the name of compounds are hardly to read- this was corrected
Line 416: „The methanol from combined fractions” – were the fraction combined?- this was corrected
Line 437/438: „were tested at the concentration range of 50 μg/mL” - „range” is unnecessary - this was corrected
Line 464: „(2 x 106)” – use the superscript for 6 - this was corrected
4.1. Chemicals and general experimental procedures – the paragraph should be divided on subsection to facilitate following – this is a good point, this was corrected
Reviewer 2 Report
In their paper Kaltun-Jasion evaluated the anti-inflammatory properties of compounds isolated from Fraxinus excelsior leaves on monocytes isolated from human peripheral blood, as model of immune effector cells. The authors provided a detailed analysis of infusion and extracts from Fraxinus excelsior leaves providing evidence of their biological activity on TNF-a, IL-6 release as well as on IL-10 receptor expression on monocytes cell surface. For their studies, authors used fractions at different percentage of methanol demonstrating the presence of ligstroside, formoside and oleoacteoside as the most represented phytocompounds. The paper sounds to be original since it gains insight for the anti-inflammatory action of leaf preparation of monocytes.
However, to consider data of this paper suitable for publication, authors should address the following aspects:
· - The expression level of TNF-a, IL-6, IL-1b as wells as that of IL-10 should be analysed by western blot analisys or Real time PCR. This approcach will allow to clarify whether exctract or compounds administered alone are capable to affect not only their release but also their expression.
· -The corresponding control of monocytes incubated in the presence of different percentage of methanol should be reported by authors to exclude any side effects.
· - At page 4, lane 157 authors describe a statistically significant effect of crude infusion and extracts, but the p value of these effects is not reported in the text. Authors should include this information.
Author Response
Thank you for your comments,
The expression level of TNF-a, IL-6, IL-1b as wells as that of IL-10 should be analysed by western blot analisys or Real time PCR. This approcach will allow to clarify whether exctract or compounds administered alone are capable to affect not only their release but also their expression.
Of course we agree. However, our study focuses on the advisability of using ash leaves in formulations to help alleviate inflammation-associated conditions. Due to the scarcity of clinical studies confirming the efficacy of this plant, our study aims to highlight its importance in phytotherapy and the need for further research into its mechanisms of action. In the next steps of our research, we are planning to expand our work to include additional techniques to elucidate the raw material's function at the molecular level, such as western-blot or PCR.
Corresponding control of monocytes incubated in the presence of different percentage of methanol should be reported by authors to exclude any side effects -opisane w wyjaśnieniu
The methanolic plant extracts were evaporated to dryness and lyophilised, then dissolved in an appropriate amount of DMSO not exceeding a total of 0.1%. To obtain the tested concentrations, the base solutions were diluted with a culture medium (RPMI 1640). The tested compounds were also dissolved in DMSO and brought to the appropriate concentration with the culture medium. The influence of 0,1% DMSO was assessed in controls.
At page 4, lane 157 authors describe a statistically significant effect of crude infusion and extracts, but the p value of these effects is not reported in the text. Authors should include this information – This was added
Round 2
Reviewer 2 Report
The authors improved their manuscript and it can be accepted for publication